# Glutamine metabolism modulates azole susceptibility in *Trypanosoma cruzi* amastigotes

Peter C Dumoulin[1], Joshua Vollrath[1,2], Sheena Shah Tomko[1], Jennifer X Wang[3], Barbara Burleigh[1]*

[1]Department of Immunology and Infectious Diseases, Harvard T.H. Chan School of Public Health, Boston, United States; [2]Institute for Pharmacy and Molecular Biotechnology, Heidelberg University, Heidelberg, Germany; [3]Harvard Center for Mass Spectrometry, Harvard University, Cambridge, United States

**Abstract** The mechanisms underlying resistance of the Chagas disease parasite, *Trypanosoma cruzi,* to current therapies are not well understood, including the role of metabolic heterogeneity. We found that limiting exogenous glutamine protects actively dividing amastigotes from ergosterol biosynthesis inhibitors (azoles), independent of parasite growth rate. The antiparasitic properties of azoles are derived from inhibition of lanosterol 14$\alpha$-demethylase (CYP51) in the endogenous sterol synthesis pathway. We find that carbons from $^{13}$C-glutamine feed into amastigote sterols and into metabolic intermediates that accumulate upon CYP51 inhibition. Incorporation of $^{13}$C-glutamine into endogenously synthesized sterols is increased with BPTES treatment, an inhibitor of host glutamine metabolism that sensitizes amastigotes to azoles. Similarly, amastigotes are re-sensitized to azoles following addition of metabolites upstream of CYP51, raising the possibility that flux through the sterol synthesis pathway is a determinant of sensitivity to azoles and highlighting the potential role for metabolic heterogeneity in recalcitrant *T. cruzi* infection.

*For correspondence:
bburleig@hsph.harvard.edu

**Competing interests:** The authors declare that no competing interests exist.

## Introduction

The goal for treatment of infectious diseases caused by pathogenic bacteria or parasites is to eliminate the pathogenic microorganism from the infected host. Pathogens that persist following treatment with an antimicrobial agent may harbor genetic mutations that give rise to resistant populations. Alternatively, the pathogen may be able to achieve a dormant, non-replicative state that becomes refractory to the treatment. A third, less explored option, is the impact of metabolic and environmental heterogeneity on the efficacy of a given antimicrobial agent (*Yang et al., 2017*). Factors such as pathogen respiration (*Lobritz et al., 2015*), ATP levels (*Conlon et al., 2016*), and buildup of metabolic intermediates (*Dumont et al., 2019*) as well as environmental stressors such as the host immune response (*Rowe et al., 2020*) can modulate antibiotic efficacy. Recent work has shown that when the metabolic state and growth rate of microbes are disentangled, the factor that correlates with antibiotic efficacy is the microbial metabolic state (*Lopatkin et al., 2019*). Similarly, standard in vitro inhibitory activity of a candidate compound can be confounded by altered pathogen metabolism due to growth media composition (*Hicks et al., 2018*; *Pethe et al., 2010*) and consequently an understanding of these interactions can potentiate treatment (*Vestergaard et al., 2017*). These complex interactions are best understood in cases of bacterial pathogenesis, but recently, similar trends are apparent in eukaryotic pathogens (*Dumont et al., 2019*; *McLean and Jacobs-Lorena, 2017*; *Murithi et al., 2020*).

A group of single-celled protozoan pathogens with significant global disease burden exhibit metabolic and growth flexibility (*Dumoulin and Burleigh, 2018*; *McConville et al., 2015*;

*Saunders et al., 2010*; *Shah-Simpson et al., 2017*) suggesting the potential for interactions with drug efficacy. The kinetoplastid protozoan parasite *Trypanosoma cruzi* is the causative agent of Chagas disease and infects approximately 6 million individuals (*WHO, 2015*) resulting in substantial morbidity (*Bern, 2015*), economic burden (*Lee et al., 2013*) and an estimated 10,000 deaths annually (*Stanaway and Roth, 2015*). Parasite transmission is most common through the triatomine insect vector but also occurs congenitally, orally and by transfusion or transplantation (*Bern et al., 2011*; *Rassi et al., 2010*). Current therapies include treatment with benznidazole or nifurtimox and include undesirable characteristics such as prolonged treatment and severe adverse events (*Castro and Diaz de Toranzo, 1988*; *Pinazo et al., 2010*; *Viotti et al., 2009*). During the chronic stages of the disease, the elimination of parasitemia (*Murcia et al., 2010*) and the clinical benefit of these therapies (*Morillo et al., 2015*; *Urbina and Docampo, 2003*) are uncertain. Since the continued presence of the parasite is the main driver of disease (*Jones et al., 1993*; *Tarleton et al., 1997*; *Zhang and Tarleton, 1999*) a central goal for new therapies is the ability to induce sterile cure.

Azole antifungal medications that target the production of endogenous sterols were promising pre-clinical candidates for Chagas disease (*Docampo et al., 1981*; *Docampo and Schmuñis, 1997*; *Lepesheva et al., 2011*; *Urbina, 1997*) due to the presence of ergostane-type sterols in *T. cruzi* and an already establish tolerability and safety profile in humans (*Zonios and Bennett, 2008*). In clinical trials, azole monotherapy resulted in parasite suppression during treatment that was not sustained following cessation of therapy (*Molina et al., 2014*; *Morillo et al., 2017*; *Torrico et al., 2018*) suggesting that parasites are sensitive to therapy even in the absence of radical cure. The inability of azoles to provide sterile cure in vivo does not appear to be due to an inferior pharmacokinetic profile or high plasma binding and suggests an additional unexplored factor may influences parasite susceptibility to these compounds (*Khare et al., 2015a*). Given the ability of intracellular *T. cruzi* amastigotes to adapt to their immediate metabolic environment (*Caradonna et al., 2013*; *Dumoulin and Burleigh, 2018*; *Shah-Simpson et al., 2017*), we sought to determine the extent to which plasticity influences parasite susceptibility to ergosterol biosynthesis inhibitors. Here, we show that glutamine metabolism modulates the ability of azoles to eliminate intracellular *T. cruzi* amastigotes, independent of growth rate.

## Results

### Exogenous glutamine levels modulate sensitivity of intracellular *T. cruzi* amastigotes to lanosterol-14α-demethylase inhibitors

The mechanisms underlying recalcitrant *T. cruzi* infection are not well understood. The spontaneous emergence of non-replicative 'latent' *T. cruzi* forms (*Sánchez-Valdéz et al., 2018*) has been put forward as a possible explanation for the failure to achieve parasitological cure following drug treatment of chronic Chagas patients. However, the potential for metabolic heterogeneity to modulate *T. cruzi* susceptibility to trypanocidal drugs has not been investigated. To begin to address this prospect, we sought to determine if variable growth conditions known to modulate the proliferative activity of intracellular *T. cruzi* amastigotes (*Dumoulin and Burleigh, 2018*) impact the susceptibility of *T. cruzi* amastigotes to trypanocidal drugs. Dose-response curves for inhibition of intracellular amastigote growth were generated for benznidazole, the first-line therapy for Chagas disease (*Bern et al., 2007*) and ketoconazole, an azole inhibitor of trypanosome sterol synthesis (*Lepesheva et al., 2011*), as outlined (*Figure 1—figure supplement 1*). Shown are the dose-response data for drug-treated cultures in medium with and without supplemental glucose or glutamine. These nutrients were included in the analysis based on knowledge that *T. cruzi* amastigotes metabolize exogenously supplied glucose and glutamine (*Shah-Simpson et al., 2017*) and that restriction of either carbon source slows amastigote replication without causing lethality (*Dumoulin and Burleigh, 2018*). Additionally, the primary neonatal human dermal fibroblasts (NHDF) used as host cells for parasite infection are tolerant of nutrient restriction (*Dumoulin and Burleigh, 2018*). Here, we find that the dose-response for benznidazole is not altered by these changes in the growth medium (*Figure 1A*). Similarly, intracellular amastigotes exposed to ketoconazole in complete medium, or medium lacking glucose, exhibited the full range of sensitivity to ketoconazole (*Figure 1B*). In contrast, inhibition of intracellular amastigote growth with increasing ketoconazole concentration was diminished when *T. cruzi*-infected monolayers were maintained in

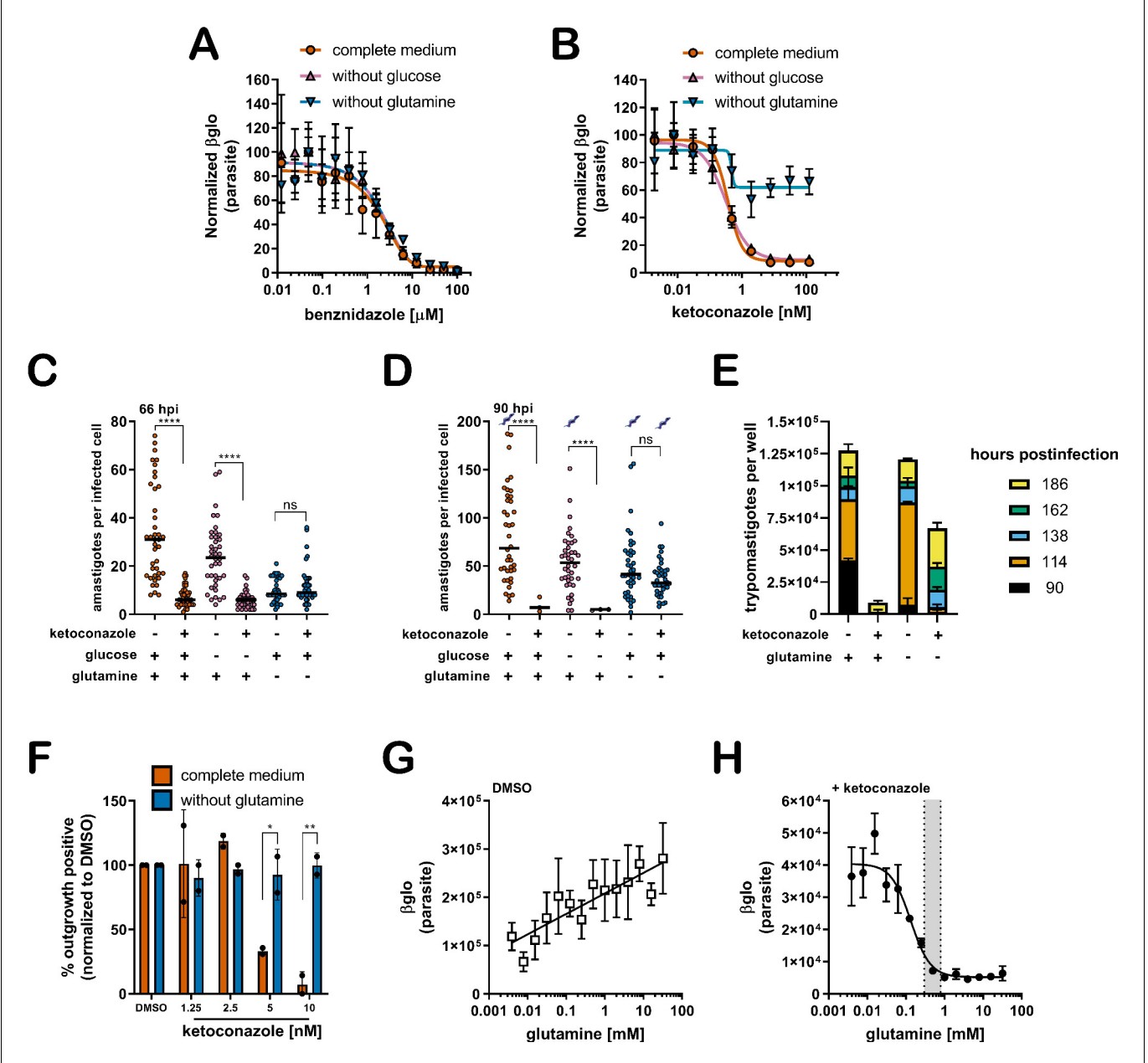

**Figure 1.** A lack of supplemental glutamine in growth medium protects intracellular *T. cruzi* amastigotes from the cytocidal effects of ketoconazole. (A) Dose response curves at 66 hpi of benznidazole and (B) ketoconazole treatment, in the indicated media compositions, normalized to the largest mean in each treatment group. Mean (symbols) and standard deviation shown (n = 4). (C) Microscopic counts at 66 hpi and (D) 90 hpi of the number of amastigotes per infected host cell (n = 40), medians indicated. Cartoons at top of graph indicate conditions where extracellular trypomastigotes are visible in the culture supernatant. (E) Growth media was replaced and extracellular trypomastigotes were counted every 24 hr beginning at 90 hpi (n = 2). (F) Detection of clonal outgrowth 14 days after the indicated treatments, normalized to DMSO (vehicle) treatment. Mean and standard deviation shown, circles indicates values of two independent experiments with 28 wells used per treatment within an experiment. (G) Dose response curves of glutamine in the presence of DMSO or (H) ketoconazole (5 nM). Mean and standard deviation shown (n = 3). Grey shading indicates in panel I shows the physiological range found in human plasma (800–300 uM) (*Cruzat et al., 2018*). Statistical comparisons between medians (C,D) were performed using a Kruskal-Wallis test with Dunn's multiple comparisons test (****p<0.0001, ns = not significant). Comparisons of means from outgrowth (F) was performed using a two-way ANOVA with Dunnett's multiple comparisons test (*p<0.05, **p<0.01).

The online version of this article includes the following figure supplement(s) for figure 1:

**Figure supplement 1.** Experimental schematic for in vitro infection and readouts.

**Figure supplement 2.** Sensitivity to additional azole drugs is modulated by glutamine.

**Figure supplement 3.** Removal of supplemental glutamine but not glucose maintains the proportion of infected host cells in the presence of azoles.

*Figure 1 continued on next page*

*Figure 1 continued*

**Figure supplement 4.** Proline or histidine supplementation do not sensitize amastigotes to ketoconazole in the absence of glutamine.

**Figure supplement 5.** Slowed amastigote growth, antioxidants, or hypoxia does not prevent the cidal effects of ketoconazole.

the absence of supplemental glutamine (*Figure 1B*). Analogous results were obtained with other azole inhibitors of lanosterol-14α-demethylase (CYP51), itraconazole, ravuconazole, and posaconazole (*Figure 1—figure supplement 2*), drugs that failed to deliver sterile cure in the clinic (*Molina et al., 2014*; *Torrico et al., 2018*) and in animal models (*Khare et al., 2015a*).

Microscopic analysis of fixed parasite-infected fibroblast monolayers confirmed these findings (*Figure 1C,D*). For these experiments, the infecting dose of *T. cruzi* trypomastigotes was titrated to achieve an average of one parasite in each infected cell prior to the introduction of a test condition (see *Figure 1—figure supplement 1*). Differences in parasite numbers per infected cell or the number of host cells that retain intracellular parasites at later time points provides a reliable metric for growth inhibition and parasite death, respectively, which cannot be resolved using the β-Glo assay (e.g. *Figure 1A,B*) a rapid higher throughput assay that reports relative parasite abundance (*Dumoulin and Burleigh, 2020*; *Dumoulin and Burleigh, 2018*). Microscopic counts reveal that growth of intracellular *T. cruzi* amastigotes exposed to 5 nM ketoconazole ($>IC_{99}$) for 48 hr (66 hpi) in complete medium or medium without glucose was significantly impaired (*Figure 1C*) as expected and with significant loss in the proportion of infected cells (*Figure 1—figure supplement 3*) suggestive of parasite death. Under conditions of glutamine restriction, intracellular *T. cruzi* amastigotes survived ketoconazole exposure (*Figure 1C*) and continued to replicate as evidenced by the greater number of amastigotes per infected cell at 90 hpi (*Figure 1D*) with no reduction in the proportion of infected cells under these conditions at this time point (*Figure 1—figure supplement 3*). The detection of extracellular trypomastigotes in the supernatants of untreated cultures and in those treated with ketoconazole in the absence of glutamine at 90 hpi (*Figure 1D*; symbols), further demonstrates that these ketoconazole-treated amastigotes are competent to complete the intracellular cycle in mammalian host cells and to produce trypomastigotes. As expected given the slower growth of amastigotes in fibroblasts cultured without glutamine and exposed to ketoconazole, the production of trypomastigotes is delayed as compared to untreated controls (*Figure 1E*).

To evaluate the longer-term impact of ketoconazole exposure on intracellular *T. cruzi* amastigotes cultured in the absence of supplemental glutamine, a clonal outgrowth assay was utilized to quantitatively measure parasite rebound following treatment (*Dumoulin and Burleigh, 2020*; *Dumoulin and Burleigh, 2018*). Detection of outgrowth (>14 days), requires surviving parasites to successfully complete several lytic cycles and therefore this assay distinguishes cytostatic from cidal effects of a test compound. Exposure of intracellular amastigotes to increasing concentrations of ketoconazole in complete medium (from 18 hpi - 66 hpi) results in a proportional decrease in clonal outgrowth (*Figure 1F*), consistent with irreversible cytotoxicity incurred by exposure to ketoconazole (*Goad et al., 1989*). In contrast, no evidence of killing was seen when supplemental glutamine was restricted during the period of ketoconazole exposure as clonal outgrowth was comparable to vehicle-treated controls under these conditions (*Figure 1F*). Extending the ketoconazole exposure time to 72 hr did not alter this outcome (not shown). Combined, these results confirm that intracellular *T. cruzi* amastigotes are protected from the lethal effects of ketoconazole when supplemental glutamine levels are restricted. Not only do the parasites survive, but they continue to replicate in the presence of 5 nM ketoconazole, which is normally lethal within the time frame of exposure. Importantly, protection occurs at a population level, as opposed to selection of a minor amastigote subpopulation that is intrinsically refractory to the drug or in a latent, non-replicative state.

## Glutamine supplementation sensitizes intracellular *T. cruzi* amastigotes to ketoconazole in a dose-dependent manner

Intracellular *T. cruzi* amastigotes succumb to the toxic effects of azoles when glutamine (2 mM) is present in the in vitro growth medium (*Figure 1A–F*). Given that standard glutamine concentrations in culture medium (1–2 mM) are significantly higher than the physiologic range of human plasma (800–300 μM) (*Cruzat et al., 2018*), supplemental glutamine was added back to glutamine-free DMSO and ketoconazole-treated cultures (*Figure 1G,H*; respectively) to determine the

concentration range of exogenous glutamine that sensitizes intracellular amastigotes to ketoconazole. In the absence of drug, the intracellular parasite load increases linearly with the addition of glutamine (*Figure 1G*), but in the presence of a fixed concentration of ketoconazole (5 nM), supplemental glutamine decreased amastigote growth in a dose-dependent manner (IC$_{50}$ of 133.4 µM for glutamine; *Figure 1H*). Addition of amino acids known to be metabolized by the epimastigote stage (proline or histidine)(*Barisón et al., 2016*; *Sylvester and Krassner, 1976*), but not present in the base mammalian growth medium, failed to sensitize amastigotes to ketoconazole or impact parasite growth in the absence of glutamine (*Figure 1—figure supplement 4*). These results point to glutamine metabolism or a glutamine sensitive process in the parasite, host cell or both, as a key factor in the susceptibility of intracellular *T. cruzi* amastigotes to azole drugs.

## Slowed parasite growth does not explain glutamine-sensitive survival of ketoconazole-treated intracellular amastigotes

Since intracellular *T. cruzi* amastigote growth is slowed under conditions of glutamine restriction in vitro (*Figure 1C,D*), and slower growing parasites may be less susceptible to inhibitors of anabolic processes such as sterol synthesis, we cannot rule out the possibility that reduced growth rate alone might protect parasites from the lethal effects of azoles. To assess whether slowed growth is an underlying factor in the protection of intracellular amastigotes from azole-mediated death, we exploited a small molecule inhibitor of parasite cytochrome b, GNF7686 (*Khare et al., 2015b*) that acts cytostatically to reduce amastigote replication rates in a dose-dependent manner and with no detectable interaction with glutamine (*Dumoulin and Burleigh, 2018*). At early time points both glutamine restriction and GNF7686 treatment protect parasites from ketoconazole (*Figure 1—figure supplement 5A,B*), but unlike the protection observed in the absence of supplemental glutamine, the slower growing GNF7686-treated parasites succumbed to ketoconazole treatment in complete medium by 90 hpi (*Figure 1—figure supplement 5C,D*). Importantly, GNF7686 treatment did not interfere with survival of azole-treated amastigotes under conditions of glutamine restriction.

Other factors associated with glutamine restriction but independent of parasite growth rate may mediate protection from azoles including the generation of reactive oxygen species (ROS) due to glutamine deprivation (*Matés et al., 2002*), cytochrome b inhibition (*Dröse and Brandt, 2008*; *Fridovich, 1978*) or variations in oxygen consumption which is a requirement for sterol synthesis (*Parks, 1978*). ROS does not play a role in protection of intracellular *T. cruzi* amastigotes from azole-mediated cytotoxicity given that antioxidant supplementation does not alter the susceptibility of amastigotes to azoles under any of the conditions tested (*Figure 1—figure supplement 5E,F*). Similar outcomes were achieved when experiments were conducted under normoxic (~20% O$_2$) or hypoxic (1.3% O$_2$) conditions (*Figure 1—figure supplement 5G–I*). Thus, combined with the observation that restriction of supplemental glucose, another amastigote growth-limiting condition fails to protect intracellular parasites from ketoconazole (*Figure 1B–D*), our results point to dysregulated glutamine metabolism, rather than slowed parasite growth or oxidative stress, in the survival of intracellular *T. cruzi* amastigotes following exposure to azoles.

## Glutamine-derived carbons are incorporated into amastigote sterols

The connection between glutamine availability and amastigote sensitivity to azoles is not clear. If CYP51 inhibition by azoles results in the build-up of 14-methylated intermediates that become toxic to the parasite, then any condition that modulates metabolic flux in the direction of CYP51 has the potential to modulate this pool of intermediates. To investigate the possibility of a metabolic link between exogenous glutamine and endogenous sterol synthesis in amastigotes, metabolic $^{13}$C-tracer analysis was performed. *T. cruzi*-infected fibroblasts were cultured in medium supplemented with universally labeled [U $^{13}$C]-glutamine (2 mM) in the presence and absence of 5 nM ketoconazole. Intracellular amastigotes were isolated at 52 hpi, a time point that maximizes labeling time and minimizes parasite loss due to azole treatment (*Figure 2—figure supplement 1*) under conditions that minimize contamination from host cell membranes (*Gazos-Lopes et al., 2017*). The GC-MS chromatograms of sterols extracted from untreated amastigotes reveal several peaks with retention times between 12.4 and 13.3 (*Figure 2A*) that disappear in ketoconazole-treated parasites (*Figure 2A*) indicating their positions up- or downstream of CYP51 in the sterol synthesis pathway (*Figure 2B*). Carbons from exogenous $^{13}$C-glutamine were incorporated into sterol species

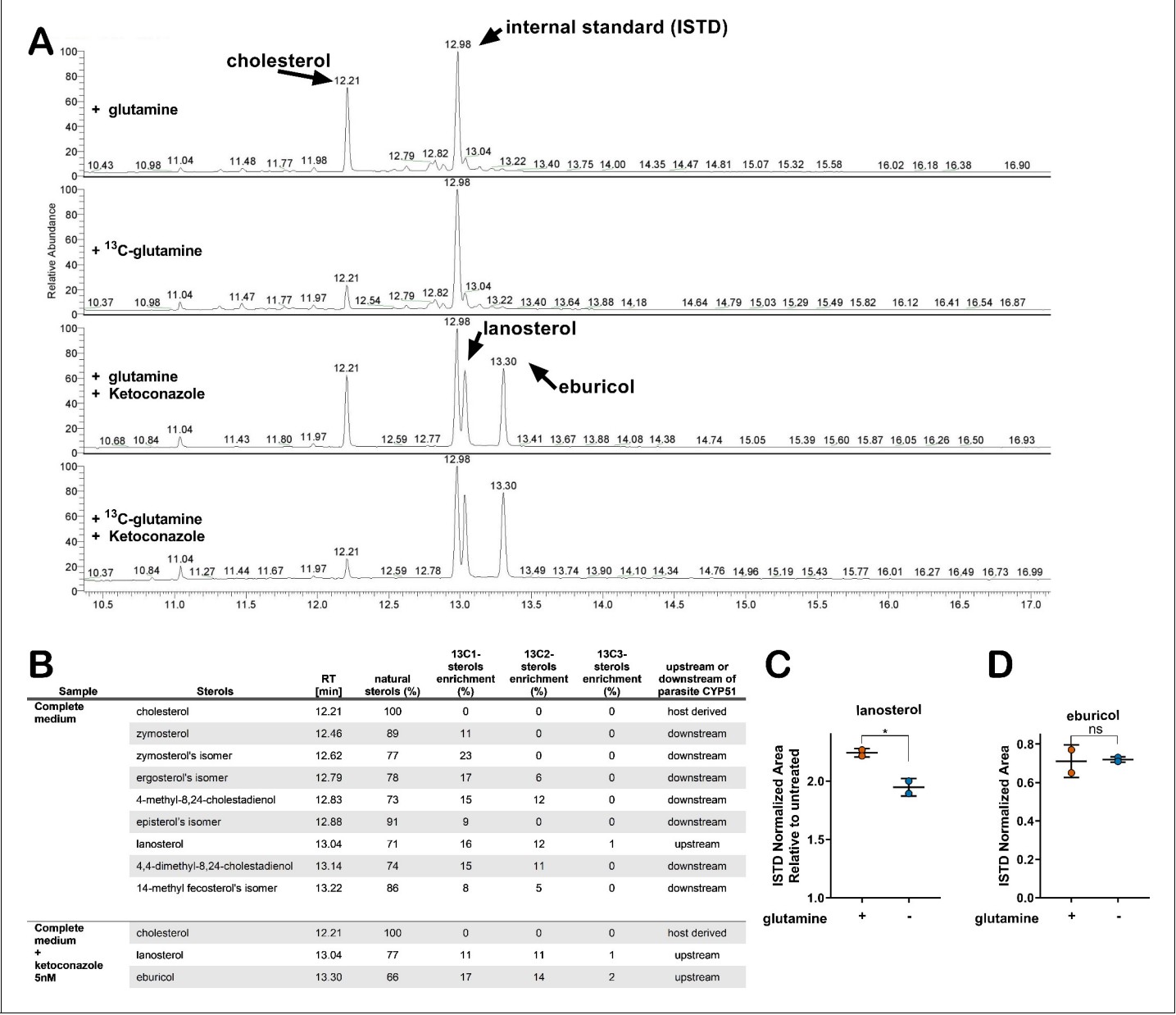

**Figure 2.** Glutamine-derived carbons are incorporated into amastigote sterols and influence the buildup of lanosterol. (**A**) Chromatogram from GC-MS detection of samples. Host-cell-derived cholesterol is seen at retention time 12.21, eburicol at 13.30, lanosterol at 13.04 and the internal standard at 12.98. (**B**) Table of detectable isolated amastigote sterol species from panel A and the percentage of natural sterols (i.e. without detectable [13]C). The proportion of species found with the indicated number of incorporated [13]C carbons are shown (e.g. 13C1, 13C2). (**C**) Quantification, using an internal standard, of lanosterol and (**D**) eburicol in isolated amastigotes (52 hpi) following treatment with ketoconazole (5 nM) at 18 hpi with or without glutamine (2 mM). Mean and standard deviation shown of independent treatments, infections and amastigote isolations (n = 2). Statistical comparisons are made using a Student's t-test (*p<0.05, ns = not significant).

The online version of this article includes the following figure supplement(s) for figure 2:

**Figure supplement 1.** Time course establishes 52 hpi as optimal time point to harvest intracellular amastigotes following ketoconazole treatment.

**Figure supplement 2.** Endogenous lanosterol and eburicol but not host derived cholesterol are reliable quantifiable from isolated intracellular amastigotes.

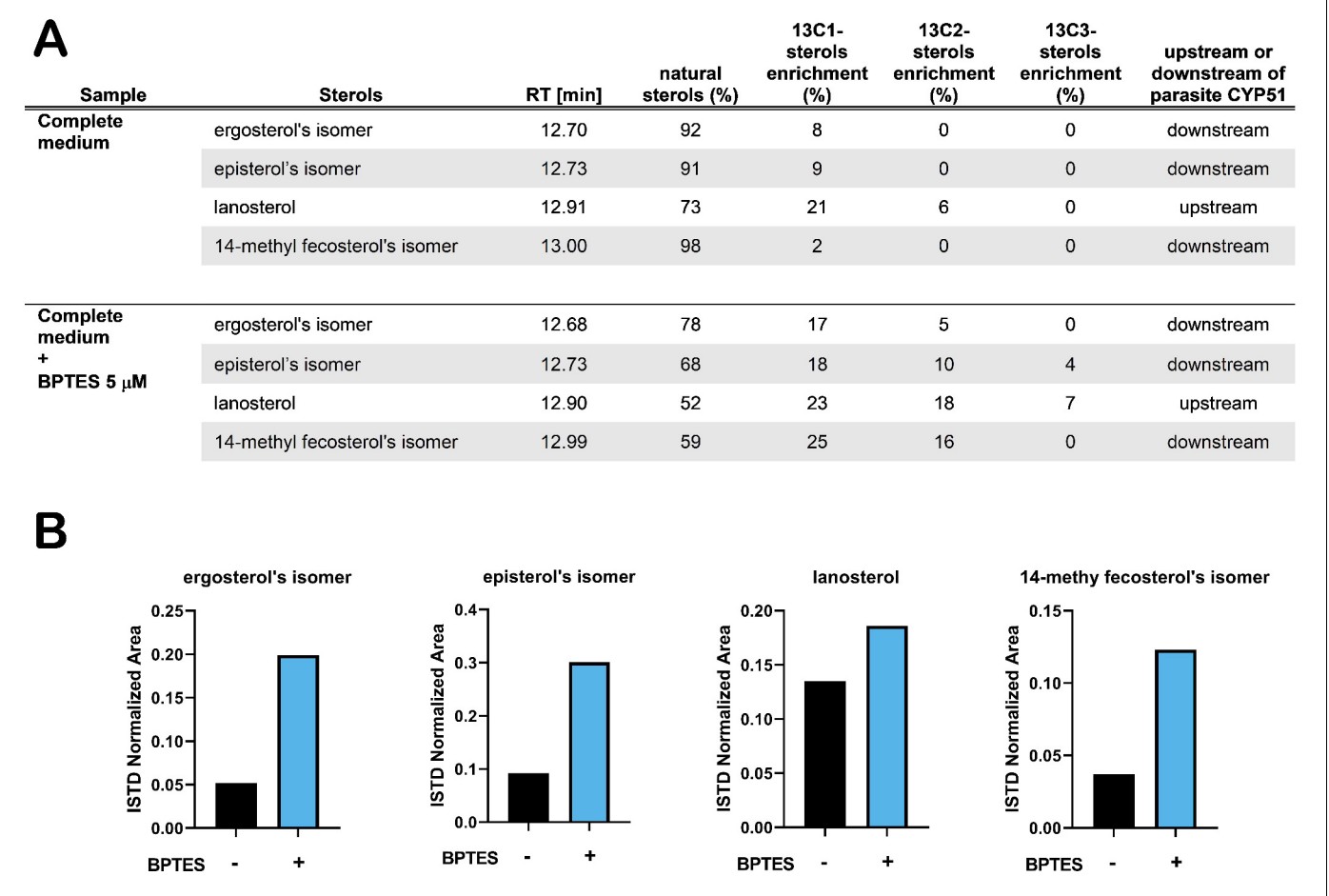

**Figure 3.** Treatment with BPTES increases incorporation of carbons from glutamine into endogenously synthesized amastigote sterols. (**A**) Table of detectable isolated amastigote sterol species from and the percentage of natural sterols (i.e. without detectable [13]C). The proportion of species found with the indicated number of incorporated [13]C carbons are shown (e.g. 13C1, 13C2). (**B**) Normalized area (ISTD) of the indicated species without and without BPTES treatment.
The online version of this article includes the following figure supplement(s) for figure 3:

**Figure supplement 1.** BPTES does not inhibit oxygen consumption of isolated amastigotes from glutamine and sensitizes amastigotes to ketoconazole.

downstream of CYP51 (e.g. zymosterol and isomers of ergosterol and episterol; *Figure 2B*). Consistent with reports that *T. cruzi* amastigotes do not generate ergosterol as a final species from endogenous sterol synthesis (*Gunatilleke et al., 2012*; *Liendo et al., 1999*; *Ottilie et al., 2017*) we could not detect canonical ergosterol in this analysis, although isomers of ergosterol were present (*Figure 2B*). [13]C-labeled lanosterol and eburicol, both upstream of CYP51, accumulated in azole-treated amastigotes (*Figure 2*) as expected (*Gunatilleke et al., 2012*; *Ottilie et al., 2017*). In addition to these sterols, host-derived cholesterol was present in all samples, independent of azole treatment and did not incorporate [13]C under any of the conditions examined. Notably, [13]C incorporation into some amastigote sterols was as high as 20–30% (*Figure 2B*), suggesting that the contribution of exogenous glutamine to the endogenous sterol pool in *T. cruzi* amastigotes is significant.

With the establishment of a metabolic link between glutamine and sterol biosynthesis in *T. cruzi* amastigotes (*Figure 2A,B*), we sought to determine the impact of glutamine restriction on amastigote sterol levels using internal standard (ISTD)-based quantification (GC-MS/ISTD). The low experimental variation in normalized lanosterol and eburicol levels (coefficient of variation <0.3; *Figure 2—figure supplement 2*) validates this comparative approach, and also highlights the variability in host-derived cholesterol in different biological samples (coefficient of variation >0.3;

*Figure 2—figure supplement 2*). We find that the increase in lanosterol that occurs with azole treatment compared to non-treated controls to be lower in amastigotes cultured in medium without supplemental glutamine (*Figure 2C*), whereas eburicol levels were similar in both conditions (*Figure 2D*).

With this reasoning, we sought to determine how perturbation of host glutamine metabolism would impact amastigote sterol pools. We employed BPTES, a small molecule inhibitor of the host glutaminase (GLS1) enzyme, to block the conversion of glutamine to glutamate in the host cell (*Robinson et al., 2007*). As *T. cruzi* lacks a glutaminase with discernable identity to human GLS1 and BPTES fails to inhibit glutamine-dependent respiration in amastigotes (*Figure 3—figure supplement 1A*), a process that requires the conversion of glutamine to glutamate, we expect that any effect(s) of BPTES are due to the inhibition of host GLS1 and related metabolic changes. Our results show that [U $^{13}$C]-glutamine incorporation into endogenously synthesized amastigote sterols is significantly increased as compared to untreated control parasites (*Figure 3A*). This result is consistent with the idea that inhibition of host GLS1 with BPTES increases glutamine availability for intracellular amastigotes to take up, increasing the flow of carbons from glutamine into the parasite sterol pool. The increase in $^{13}$C-labeling of amastigote sterols in the presence of BPTES coincides with a marked increase in the relative amounts of endogenously synthesized sterols (*Figure 3B*), bolstering the conclusion that sterol production in intracellular *T. cruzi* amastigotes is influenced by glutamine availability and/or flux through the sterol synthesis pathway.

## Treatment with BPTES or supplementation with pathway intermediates is sufficient to re-sensitize *T. cruzi* amastigotes to ketoconazole in the absence of glutamine

The generation of 14-methylated sterol precursors has been implicated in the detrimental phenotypes associated with inactivation of CYP51 in other kinetoplastid protozoan parasites and in yeast (*Goad et al., 1989*; *Kelly et al., 1995*; *Mukherjee et al., 2019*). If flux and/or generation of these methylated intermediate species modulates the sensitivity of *T. cruzi* amastigotes to azoles, we reasoned that BPTES treatment, which results in increased incorporation of carbons from glutamine into parasite sterols (*Figure 4A*), could re-sensitize the parasites to ketoconazole under conditions of glutamine restriction. While BPTES treatment alone had no measurable effect on *T. cruzi* amastigote replication, parasites succumbed to ketoconazole treatment in the absence of glutamine when BPTES was present (*Figure 3—figure supplement 1B*) suggesting that glutamine-dependent sensitization of intracellular parasites to azoles is correlated with increased flux through the sterol synthesis pathway.

Next, we sought to determine if provision of metabolites downstream of glutamine in this pathway, but upstream of CYP51 (*Figure 4A*), would have a similar effect in increasing the susceptibility of intracellular amastigotes to ketoconazole in the absence of supplemental glutamine. Similar to the results with BPTES (*Figure 4B*), addition of a cell-permeable form of α-KG, dimethyl α-ketoglutarate (di-α-KG), to parasite-infected cells resulted in a significant reduction of intracellular amastigote growth following treatment with ketoconazole in the absence of glutamine (*Figure 4C*). However, as compared to the results with BPTES, the effect of di-α-KG was less dramatic and may reflect conversion of di-α-KG to glutamate and then to glutamine.

Moving down the pathway, we examined the possibility that delivery of an isoprenoid precursor of sterol synthesis, farnesol pyrophosphate (FPP), or farnesol itself would have a similar impact on amastigote sensitivity to ketoconazole in the absence of glutamine. In *T. cruzi*, isoprenoid precursors can enter the endogenous sterol synthesis pathway (*Cosentino and Agüero, 2014*) and in other systems, exogenous supplementation of isoprenoid precursors is sufficient to chemically rescue blockage of essential metabolic function (*Yeh and DeRisi, 2011*). In the absence of ketoconazole, addition of farnesyl pyrophosphate (FPP) (*Figure 4D*) or farnesol (*Figure 4E*) to the culture medium had no effect on amastigote growth. In contrast, intracellular amastigotes failed to survive ketoconazole treatment in glutamine-free medium when FPP or farnesol were present (*Figure 4—figure supplement 1*). Although we cannot rule out the possibility that the sensitizing effects of BPTES, or exogenous di-α-KG, FPP or farnesol are due to mechanisms not directly involved with endogenous sterol synthesis, these results are consistent with the conclusion that modulation of a pathway from glutamine to sterol production in intracellular *T. cruzi* amastigotes has a profound impact on the ability of this parasite to survive the lethal effects of azoles. As such, our findings highlight glutamine

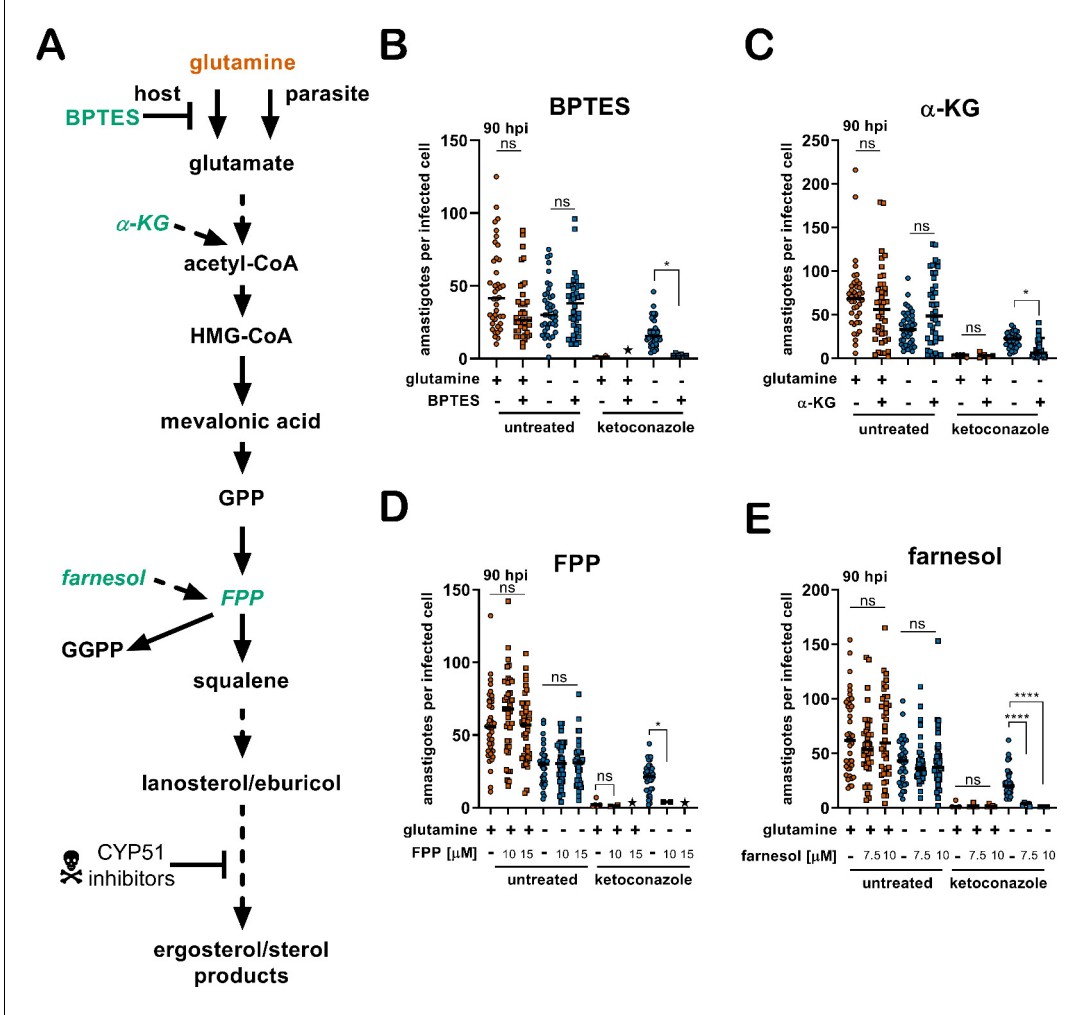

**Figure 4.** Addition of metabolites re-sensitize intracellular *T. cruzi* amastigotes to ketoconazole in the absence of glutamine. (A) Schematic of endogenous sterol synthesis. Dash lined arrows indicate omission of steps for simplicity. (B) Microscopic counts of amastigotes per infected cell (n = 40) at 90 hpi treated with BPTES (5 µM), (C) αKG (10 mM), (D) FPP or (E) farnesol. Statistical comparisons between medians were performed using a Kruskal-Wallis test with Dunn's multiple comparisons test (****p<0.0001, ***p<0.001, *p<0.05, ns = not significant).

The online version of this article includes the following figure supplement(s) for figure 4:

**Figure supplement 1.** Addition of metabolites re-sensitize infected cultures to ketoconazole in the absence of glutamine.

metabolism as a process that has the potential to be manipulated to improve the efficacy of antifungal azoles against *T. cruzi*.

## Discussion

The metabolic state of a microorganism can be influenced by its immediate environment and have an impact on the efficacy of antimicrobials, independent of growth rate (*Lopatkin et al., 2019*). For pathogenic microbes, this environment is largely dependent on the status of its host. Within a host, nutrient utilization and availability vary widely across tissues (*Aichler and Walch, 2015*; *Shlomi et al., 2008*). Even within a single tissue, the presence of an inflammatory response alters local metabolism (*Kominsky et al., 2010*) and in many cases leads to intracellular nutrient restriction to control pathogen growth (*Grohmann et al., 2017*). Pathogen growth in vitro cannot always reflect the complete spectrum of metabolic environments present in vivo and consequently can confound interpretations of standard antimicrobial assays (*Hicks et al., 2018*; *Pethe et al., 2010*). These considerations may be especially pertinent to *T. cruzi*, a parasite that in its mammalian host replicates

intracellularly in diverse tissues and persists for the lifetime of the host, exposing the parasite to an immune response that suppresses parasitemia without sterile cure (*Lewis et al., 2015*).

Recent clinical trials investigating the efficacy of azoles (CYP51 inhibitors) to eliminate *T. cruzi* parasitemia resulted in an initial elimination of peripheral parasitemia that, unlike benznidazole the first-line therapeutic, was not maintained after cessation of therapy (*Molina et al., 2014*; *Morillo et al., 2017*; *Torrico et al., 2018*). The anti-parasitic activity of azoles without sterile cure suggests the possibility that heterogeneous environments and/or distinct populations of parasites within a single host may underlie treatment failure. We discovered that the response of intracellular *T. cruzi* amastigotes to azoles was significantly impacted by the concentration of supplemental glutamine in the growth medium. Rather than manifesting as a shift in traditionally measured $IC_{50}$, this effect was characterized by the inability of azoles to cause radical growth reduction at all azole concentrations tested under conditions of low or no supplemental glutamine. This observation demonstrates a novel link between metabolism and drug efficacy against *T. cruzi* that is specific to the nature of the treatment because we find that the efficacy of benznidazole is unchanged in these conditions.

The activity of ketoconazole is derived from blockage of an anabolic process (i.e. sterol synthesis) that is potentially less active in more slowly diving parasites. Since glutamine restriction slows amastigote proliferation we investigated if slowed growth in general, rather than glutamine metabolism, explains the observed protection. We demonstrated that slowing amastigote growth using an inhibitor (GNF7686) of parasite cytochrome b delayed, but did not prevent amastigote death due to azoles. In a similar scenario, more slowly growing *T. cruzi* isolates have been shown to be less susceptible to azoles in a single time point growth inhibition assay, yet rapidly dividing strains can still outgrow following treatment (*MacLean et al., 2018*). Taken together, these data show that the dynamics of azole mediated killing of replicating amastigotes is influenced by parasite growth rate, but that growth rate is insufficient to explain the protection mediated by glutamine withdrawal shown here. An alternative explanation for parasite persistence in vivo is a cessation of amastigote division. While the nature of dormancy in *T. cruzi* remains under investigation (*Sánchez-Valdéz et al., 2018*), we report here a protective mechanism that allows for continued amastigote proliferation in the presence of drug at the population level. Since, slowed growth appears to induce limited tolerance to azoles but is insufficient to provide resistance; we investigated potential mechanisms to explain the protection from azoles mediated specifically by glutamine restriction.

Similar to fungal species, *T. cruzi* endogenously synthesizes ergostane-type sterols. As parasites replicate, the consequences of azoles that block ergostane-type sterol synthesis are two-fold: a gradual depletion of sterol end products and the buildup of 14-methylated sterol synthesis intermediates. Drug resistance to azoles observed in fungal pathogenesis is mediated by alleviating drug activity or mitigating the consequences of azole activity. These mechanisms include mutations in the target CYP51 (*Howard et al., 2009*), drug efflux (*Prasad and Rawal, 2014*), selection for sterol auxotrophy (*Hazen et al., 2005*) or suppressor mutations that alter the composition of 14-methylated sterol synthesis intermediates (*Kelly et al., 1995*). Target site or suppressor mutations cannot explain protection mediated by glutamine restriction in *T. cruzi* amastigotes in our study because we found that amastigotes are protected as a population, which occurs rapidly within a single lytic cycle. Under these conditions, amastigotes are not exposed to prolonged selection. Protection from azoles mediated by glutamine withdrawal is likely not due to a decrease in activity due to drug efflux, because the generation of sterols downstream of CYP51 is abolished in the absence of glutamine, demonstrating that the activity of ketoconazole is unchanged. These data demonstrate that protection may not be mediated by changes to the activity or sensitivity of CYP51 to azoles but rather changes to the consequences of CYP51 inhibition.

Inhibition of CYP51 results in a buildup of 14-methylated sterols and an absence of downstream sterols. In another kinetoplastid parasite, increased membrane fluidity and heat sensitivity are seen in *Leishmania major* CYP51 knockouts (*Xu et al., 2014*) but not in knockouts of sterol methyltransferase (*Mukherjee et al., 2019*) suggesting that the accumulation of 14-methylated sterols rather than the absence in ergosterol effects parasite viability. Similarly, 14-methylated sterols are found in *T. cruzi* amastigotes when treated with azoles. We found that the carbons from glutamine enter the endogenous sterol synthesis pathway in *T. cruzi* amastigotes and are incorporated into the 14-methylated sterol synthesis intermediates lanosterol and eburicol. The incorporation of these carbons into amastigote sterols indirectly suggests that removal of glutamine has the potential to diminish flux through the sterol synthesis pathway. Although both lanosterol and eburicol increase in the presence

of ketoconazole, the relative amount of lanosterol is less when amastigotes are grown in the absence of glutamine, suggesting that increased amounts of 14-methylated sterols influence susceptibility to azoles. As we have only measured free sterols in isolated *T. cruzi* amastigotes, it is possible that additional sterols or their synthesis intermediates are esterified (*Pereira et al., 2018*; *Taylor and Parks, 1978*) or exported from the amastigote and therefore not detected using these methods. Additionally, we have not measured changes in cholesterol scavenging from the host cell by the parasite. It is still unclear if *T. cruzi* amastigotes balance their endogenous synthesis of sterols with exogenous scavenged cholesterol from the host and how this dynamic can alter azole mediated killing.

An increase in 14-methylated sterols directly upstream of CYP51 during azole treatment demonstrates that metabolites are still committed to this pathway even when CYP51 is inhibited. If flux through the endogenous sterol synthesis pathway is insufficiently regulated in the presence of azoles, and a buildup of 14-methylated sterols is associated with susceptibility to azoles we reasoned that altering flux into this pathway may potentiate the activity of azoles, even in the absence of glutamine. In line with this hypothesis, when BPTES is used to block the metabolism of glutamine by the host cell, we find an increase in both incorporation of glutamine-derived carbons into parasite sterols and an increase in their overall abundance. In addition to altering incorporation of glutamine into sterols, BPTES concurrently sensitizes amastigotes to ketoconazole. Since BPTES acts early in glutamine metabolism we cannot formally exclude the possibility that other glutamine fueled pathways contribute to sensitization. Additionally, the intermediate steps of glutamine metabolism into parasite sterols remain obscure since the incorporation of labeled carbons into sterols do not occur at proportions observed in other systems (*Metallo et al., 2012*). In support of glutamine metabolism influencing amastigote sensitivity to azoles, the addition of metabolites (α-KG/farnesol/FPP) also re-sensitive amastigotes to azoles in the absence of glutamine. However, re-sensitization to ketoconazole by farnesol/FPP may be mediated through an alternative pathway (e.g. prenylation) or through direct alterations to amastigote membranes that may be compromised by a reduction in ergostane-type sterol end products and the presence of free 14-methylated sterol species. Taken together these data show that carbons derived from glutamine enter the endogenous sterol synthesis pathway in amastigotes and that changes (direct or indirect) in sterol synthesis flux may influence the buildup of 14-methylated species and azoles sensitivity.

While glutamine is the most abundant amino acid in the human body it has a wide intracellular distribution between tissues (*Cruzat et al., 2018*) and as a result proliferating amastigotes are predicted to be exposed to a variety of glutamine levels in vivo. Standard in vitro growth media compositions contain supraphysiologic amounts of glutamine to allow for the sustained growth of rapidly dividing cells. Data from this study show that in vitro growth conditions may belie the variable efficacy of candidate anti-parasitic compounds and offer complementary approaches to better prioritize new candidates.

The novel observations presented have implications for *T. cruzi* antimicrobial prioritization and further evidence that the metabolic state of a microorganism is an important consideration for determining drug susceptibility. Even though the identification of new targets for antiparasitic compounds (*Khare et al., 2016*; *Khare et al., 2015b*) is promising, a better understanding of parasite metabolism and reasons for failure of prior candidates has the potential to aid in the prioritization of these potential therapies. In addition, the ability to modulate drug susceptibility through nutrient availability in vitro suggests that nutrient supplementation in vivo should be explored as a potential combination therapy.

## Materials and methods

**Key resources table**

| Reagent type (species) or resource | Designation | Source or reference | Identifiers | Additional information |
|---|---|---|---|---|
| Strain (*Trypanosoma cruzi*) | Tula-βgal | ATCC | PRA-330 | Tulahuén LacZ clone C4: PMID:8913471 |
| Cell line (*Macaca mulatta*) | LLC-MK2 | ATCC | CCL-7 | PMID:14449902/14449901 |

*Continued on next page*

*Continued*

| Reagent type (species) or resource | Designation | Source or reference | Identifiers | Additional information |
| --- | --- | --- | --- | --- |
| Cell line (*Homo sapiens*) | NHDF | Lonza | CC-2509 | Normal Human Neonatal Dermal Fibroblasts |
| Chemical compound, drug | ketoconazole | Enzo | Cat# EI107 | ≥99% (HPLC) |
| Chemical compound, drug | GNF7686 | Vitas-M Laboratory | Cat# STK393240 | PMID:26186534 |
| Other | glutamine | Gibco | Cat# A2916801 | |
| Other | $^{13}$C-glutamine | Cambridge Isotope Laboratories, Inc | Cat# CLM-1822 | Chemical Purity 98% |

## Cell lines

Tulahuén LacZ clone C4 (Tula-βgal), PRA-330 (ATCC, Manassas, Virginia) and LLC-MK$_2$, CCL-7 (ATCC, Manassas, Virginia) cells were obtained directly from ATCC. Normal Human Neonatal Dermal Fibroblasts (NHDF; Lonza, Basel, Switzerland) were obtained directly from Lonza, catalog number CC-2509. Testing for mycoplasma contamination was performed monthly using the PlasmoTest (InvivoGen, San Diego, California) HEK-Blue-2 kit.

## Mammalian cell culture

Mammalian cells were maintained at 37°C in a 5% CO$_2$ incubator. Dulbecco's modified Eagle medium (DMEM; HyClone, Logan, Utah) supplemented with 10% FBS (Gibco, Waltham, Massachusetts), 25 mM glucose, 2 mM L-glutamine, and 100 U/mL penicillin-streptomycin was used for propagated for uninfected cultures (DMEM-10). Unless stated otherwise, cultures infected with *Trypanosoma cruzi* were maintained in DMEM with 2% FBS (DMEM-2). Normal Human Neonatal Dermal Fibroblasts (NHDF; Lonza, Basel, Switzerland) were passaged prior to reaching confluence.

## Parasite maintenance

Tula-βgal, (ATCC, Manassas, Virginia) was passaged weekly in LLC-MK$_2$ (ATCC, Manassas, Virginia) cells (*Buckner et al., 1996*). Trypomastigotes were prepared by collecting the supernatant from infected cultures and centrifuging for 10 min at 2060 x g followed by incubation at 37°C for >2 hr to allow for trypomastigotes to swim from the pellet. After incubation the supernatant containing trypomastigotes was collected and washed in DMEM-2, enumerated using a Neubauer chamber and used for subsequent infections.

## Quantification of parasite load by luminescence

Tula-βgal parasite load was measured using luminescence as described previously (*Caradonna et al., 2013*; *Shah-Simpson et al., 2017*). One day prior to infection NHDFs were seeded in 384-well plates (Corning, Corning, New York) at a density of 1,500 cells per well and allowed to attach. Purified trypomastigotes were added at a multiplicity of infection (MOI) of 1.25 and allowed to invade for 2 hr, followed by two washes with PBS and subsequent addition of DMEM-2 without phenol red. Treatments were initiated at 18 hr post infection (hpi) to avoid any potential impacts of trypomastigote invasion and/or differentiation. At the indicated time points, growth media was removed and 10 µl Beta-Glo (Promega, Madison, Wisconsin) was added per well. Plates were incubated for >30 min at room-temperature to allow the reaction to reach equilibrium and read using an EnVision plate reader (PerkinElmer, Waltham, Massachusetts). Luminescence from uninfected wells was determined for each treatment and subtracted from infected wells to account for signal not derived from parasites.

## Compound and supplement stocks

Compounds were purchased and diluted to stock concentrations: Ketoconazole (Enzo, Farmingdale, New York) 15 mM stock in DMSO, Ravuconazole (Sigma, St. Louis, Missouri) 15 mM DMSO, Itraconazole (BioVision, Milpitas, California) 15 mM DMSO, GNF7686 (Vitas-M Laboratory, Champaign,

Illinois) 5 mM stock in DMSO, FPP (Sigma, St. Louis, Missouri) 2.3 mM stock in methanol, Farnesol (Sigma, St. Louis, Missouri) 100 mM in ethanol, NAC (Sigma, St. Louis, Missouri) 200 mM in DMEM base, Glutathione (Sigma, St. Louis, Missouri) 162 mM in media, benznidazole (Sigma, St. Louis Missouri) 20 mM in DMSO, BPTES (Selleckchem, Houston Texas) 20 mM in DMSO.

## Microscopy

Host cells were seeded 1 day prior to infection on coverslips (EMS, Hatfield, Pennsylvania) in 24-well plates at a density of $4 \times 10^4$ cells per well. Cells were infected for 2 hr at a MOI of 2 and subsequently washed twice with PBS followed by addition of DMEM-2. Coverslips were fixed in 1% PFA-PBS and stained in a 0.1% Triton X-100–PBS solution containing 100 ng/ml DAPI (Sigma, St. Louis, Missouri) for 5 min. After staining, coverslips were washed with PBS and mounted with ProLong Anti-fade (Thermo Fisher, Waltham, Massachusetts) on glass slides. Amastigotes were counted using a Nikon eclipse TE300. Amastigotes per infected host cell and the number of infected host cells per microscopic field were recorded.

## Western blot

Uninfected cells were lysed in 1 mL M-PER Mammalian Protein Extraction Reagent (Thermo Fisher, Waltham, Massachusetts) directly in culture wells and boiled for 10 min. Soluble lysate (50 µg) was loaded onto a 10% Mini-Protean TGX Gel (Bio-Rad, Hercules, CA). Proteins were transferred to a nitrocellulose membrane and blocked with a 1:1 dilution of SEA BLOCK (Thermo Fisher, Waltham, Massachusetts): PBS overnight at 4°C. The membrane was probed in blocking buffer with anti-Hif1a EPR16897 (1:1500) (Abcam, Cambridge, MA) and anti-βactin (Sigma, St. Louis, Missouri) (1:1000) for 1 hr at room temperature in hybridization tubes. After probing the membrane was washed in 1X PBS for 30 min, replacing PBS every 5 min for a total of 6 washes. Secondary antibodies, anti-mouse DyLight 680 (Cell Signaling, Dancers, MA) (1:15,000) and anti-rabbit Dylight 800 (Thermo Fisher, Waltham, Massachusetts) (1:10,000) were added and incubated for 1 hr at room temperature. The membrane was visualized using a LI-COR imaging system (LI-COR, Lincoln, NE).

## Sterol extraction

The method for extraction of sterols was based on protocols described in *Sharma et al., 2017*. Extraction occurred in glass PYREX tubes (Corning, Corning, New York) and all solvents used were HPLC grade or higher. Lipids were first extracted three times from cell pellets using C:M (2:1, v/v) and centrifuged each time at 1800 x g for 15 min at 4°C followed by collection of the supernatant in new tubes. The supernatant was dried under a constant stream of $N_2$ and the resulting material was subjected to a Folch's partitioning (4:2:1.5, C:M:W). The lower phase was removed, dried under $N_2$ and re-suspended in chloroform, passed over a silica 60 column and eluted with chloroform.

## GC-MS

GC/MS analysis was performed on a Thermo Scientific TRACE 1310 Gas Chromatograph equipped with a Thermo Scientific Q Exactive Orbitrap mass spectrometry system. Fifty microliters of the (BSTFA+10% TMCS)/pyridine (5/1 v/v) was added into each vial, vortexed well, and heated at 70°C for 30 min. A total of 1 µL sample was injected into a Thermo fused-silica capillary column of cross-linked TG-5SILMS (30 m x 0.25 mm x 0.25 µm). The GC conditions were as follows: inlet and transfer line temperatures, 290°C; oven temperature program, 50°C for 0 min, 24°C/min to 325°C for 5.7 min; inlet helium carrier gas flow rate, 1 mL/min; split ratio, 5. The electron impact (EI)-MS conditions were as follows: ion source temperature, 310°C; full scan m/z range, 30–750 Da; resolution, 60,000; AGC target, 1e6; maximum IT, 200 ms. Data were acquired and analyzed with Thermo TraceFinder 4.1 software package. Standards for cholesterol, ergosterol, lanosterol, episterol, and zymosterol were used for identification. Universal $^{13}$C-glutamine was re-suspended to a stock concentration of 200 mM in water (Cambridge Isotope Labs, Tewksbury, Massachusetts). Prior to sterol extraction, sitosterol-d7 (Avanti Polar Lipids, Alabaster, Alabama) was added as an internal standard (ISTD) at 1.12 µg/2e7 isolated amastigotes. Thermo Fisher Scientific's data analysis software Compound Discoverer 3.1 was used for the measurement of the enrichment of 13C-sterols, with 30 being the maximum number of exchangeable carbon atoms.

## Amastigote isolation

Infected monolayers were washed two times with PBS and cell detachment was achieved using a sterol free dissociation reagent, Accumax (Innovative Cell Technologies, San Diego, California). Cell suspensions were washed two times with PBS by centrifugation at 700 x g for 10 min at 4°C. The resulting cell pellets were lysed by passage through a 28-gauge needle or using the Miltenyi Gentle-MACS dissociator (M tubes, Protein_01 protocol). Lysate was passed over a PD-10 column (GE Healthcare, Chicago, Illinois) equilibrated with PBS. Eluted parasites were washed three times in PBS by centrifugation at 2300 x g at 4°C.

## Clonal outgrowth

Measurement of clonal outgrowth utilized a modified protocol from *Dumoulin and Burleigh, 2018* to allow for detection by luminescence. Host cells were seeded in 384 well plates and 25 trypomastigotes per well were allowed to invade for 2 hr, followed by two washes with PBS to removed uninvaded trypomastigotes. Treatments were initiated at 18 hpi and wells were washed at 66 hpi twice with PBS followed by addition of DMEM-2. Cultures were allowed to grow for 14 days and subsequently measured for presence of parasites by luminescence as described previously.

## Seahorse bioenergetics profiling

Amastigotes were isolated from infected cultures and prepared as described (*Shah-Simpson et al., 2016*). Isolated amastigotes were profiled in Krebs-Henseleit Buffer (KHB) with 2 mM glutamine as the sole carbon source and received either BPTES (5 µM) and Antimycin A (1 µM) or KHB + glutamine during the assay. BPTES was injected three times, bringing the concentration in the well from 0 µM at baseline to 5 µM after the first injection, 10 µM after the second injection, and 15 µM after the third injection. 1 µM Antimycin A (AA) was injected last to indicate how much of the oxygen consumption rate (OCR) was due to non-mitochondrial respiration vs. basal respiration. Measurements were done with 1 min of mixing, 1 min of waiting, and 2 min of measuring.

## Acknowledgements

We acknowledge the ICCB-Longwood Screening Facility at Harvard Medical School for help with optimization of plate-based luminescence assays. We also thank Dr. Igor C Almeida and Dr. Lucas Pagura for help with sterol extraction protocols. This work was funded by NIH NIAID R21 AI146815-01 awarded to BAB, American Heart Association Founders Affiliate Postdoctoral fellowship 19POST34380209 awarded to PCD, a Deutscher Akademischer Austauschdienst (DAAD) Programm zur Steigerung der Mobilität von Studierenden deutscher Hochschulen (PROMOS) fellowship awarded to JV, and McLennan Family Challenge Grant Program, Harvard TH Chan School of Public Health Dean's Fund for Scientific Advancement awarded to BAB.

## Additional information

### Funding

| Funder | Grant reference number | Author |
|--------|------------------------|--------|
| National Institutes of Health | R21AI146815 | Barbara Burleigh |
| American Heart Association | 19POST34380209 | Peter C Dumoulin |
| Harvard T.H. Chan School of Public Health | McLennan Family Challenge Grant Program | Barbara Burleigh |
| Deutscher Akademischer Austauschdienst | PROMOS | Joshua Vollrath |

The funders had no role in study design, data collection and interpretation, or the decision to submit the work for publication.

## Author contributions
Peter C Dumoulin, Conceptualization, Formal analysis, Supervision, Funding acquisition, Investigation, Methodology, Writing - original draft, Writing - review and editing; Joshua Vollrath, Sheena Shah Tomko, Investigation, Writing - review and editing; Jennifer X Wang, Investigation, Methodology, Writing - review and editing; Barbara Burleigh, Conceptualization, Supervision, Funding acquisition, Writing - original draft, Writing - review and editing

## Author ORCIDs
Peter C Dumoulin (iD) https://orcid.org/0000-0003-4377-0964
Joshua Vollrath (iD) http://orcid.org/0000-0002-6673-3050
Barbara Burleigh (iD) https://orcid.org/0000-0002-3642-7247

## Decision letter and Author response
Decision letter https://doi.org/10.7554/eLife.60226.sa1
Author response https://doi.org/10.7554/eLife.60226.sa2

# Additional files
## Supplementary files
• Transparent reporting form

## Data availability
Data generated or analysed during this study are included in the manuscript and supporting files.

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
