## [Decision Letter]

**Acceptance summary:**

This work demonstrates how the metabolic state of the intracellular pathogen, *Trypanosoma cruzi* directly influences the efficacy of clinically important azole drugs. The sensitivity of intracellular parasite stages to azoles increases markedly in the presence of physiologically relevant concentrations of glutamine, independent of growth rate. Metabolic labeling and inhibitor studies show that glutamine metabolism leads to enhanced sterol synthesis with concomitant accumulation of aberrant sterols and cytotoxicity in the presence of azoles. These findings suggest that metabolic heterogeneity in pathogenic protists, as in bacteria, contribute to the failure of some drugs to achieve sterile cure.

**Decision letter after peer review:**

Thank you for submitting your article "Glutamine metabolism modulates azole susceptibility in *Trypanosoma cruzi*amastigotes" for consideration by *eLife*. Your article has been reviewed by three peer reviewers, including Malcolm J McConville as the Reviewing Editor and Reviewer #1, and the evaluation has been overseen by Dominique Soldati-Favre as the Senior Editor. The following individual involved in review of your submission has agreed to reveal their identity: Kai Zhang (Reviewer #2).

The reviewers have discussed the reviews with one another and the Reviewing Editor has drafted this decision to help you prepare a revised submission.

Our expectation is that the authors will eventually carry out the additional experiments and report on how they affect the relevant conclusions either in a preprint on bioRxiv or medRxiv, or if appropriate, as a Research Advance in *eLife*, either of which would be linked to the original paper

Summary:

In this study, Dumoulin et al. show that *Trypanosoma cruzi*amastigotes are sensitized to azoles, such as ketoconazole, when exposed to increasing concentrations of glutamine, but not glucose, in their intracellular niche. They present evidence that sensitization is due to increased carbon flux into ergosterol biosynthesis, rather than changes in parasite growth rate. The findings suggest that variability in external nutrient levels and metabolic reprogramming contributes to non-genetically encoded drug resistance mechanisms in these pathogens and non-sterile cure in vivo.

Essential revisions:

The reviewers appreciated the quality of the work and the broader implications of the findings. However, a number of issues were raised that should either be addressed with addition experimental evidence or suitable caveats included in the text and/or relevant data removed from the manuscript.

1) The authors provide strong evidence that glutamine restriction alleviates the build-up of toxic 14-methyl sterol intermediate induced by ketoconazole by showing that amastigotes are re-sensitized by addition of upstream metabolites of CYP51 (like FPP). However, the phenotype of the mGDH-deficient *T. cruzi*amastigotes does not support the hypothesis that increased glutamate flux is responsible for sensitization to ketoconazole. Specifically, one would expect that the mutant line should be more resistant to ketoconazole in the presence of glutamine, which is not observed. The authors should reconcile these results with their model. It was noted that *T. cruzi* may express another active GDH which could compensate for loss of the deleted GDH, or alternatively that the GDH reaction might be by-passed by transaminase reactions. Further experiments may be needed to define the role of these GDH enzymes in ketoconazole sensitivity (including testing the effect of GDH overexpression if effective inhibition of glutamate catabolism is not possible), or these data removed from the manuscript. Along the same lines, further discussion on the phenotype of the cIDH mutant (what was expected and what was observed) is warranted.

2) Further explanation of the 13C-glutamine labelling experiments is needed (Figure 3B). The major sterol isotopomers detected in the GC-MS analyses contain +1 and +2 13C-atoms. One would expect +2 or higher isotopomers (i.e. +4, +6 etc) if glutamine is used to synthesize acetyl-CoA via citrate (either by oxidative or reductive pathways). How do the authors reconcile the observed labelling of the sterols and/or have they considered whether fatty acid b-oxidation might also be contributing to elevated acetyl-CoA and sterol synthesis under glucose limiting conditions ?

3) The evidence for the accumulation of 14-methylated sterol intermediates after ketoconazole treatment is not clear (Figure 3). Specifically, the authors should address the following issues.

• Is there any verification to show amastigote purity?

• Figure 3A: the TLC shows a loss of ergosterol/ergosterol isomer after ketoconazole treatment but no accumulation of CYP51 substrates is detected. Is the loading equal?

• Figure 3B: it seems that cholesterol is the most abundance sterol species in amastigotes based on natural sterol %, but it is not consistent with Figure 3C.

• Resolution for Figure 3B-C is insufficient.

• Figure 3C: what is the most dominant peak at 12.8-13.0 in all samples? Is it lanosterol or ergosterol isomer?

4) Figure 4: while the presented data is compelling, evidence is missing that these isoprenoid precursors are indeed affecting sterol production, rather than acting via a tertiary mechanism. Such evidence should be provided, or conclusions in text tempered.

---

## [Author Response]

Essential revisions:The reviewers appreciated the quality of the work and the broader implications of the findings. However, a number of issues were raised that should either be addressed with addition experimental evidence or suitable caveats included in the text and/or relevant data removed from the manuscript.1) The authors provide strong evidence that glutamine restriction alleviates the build-up of toxic 14-methyl sterol intermediate induced by ketoconazole by showing that amastigotes are re-sensitized by addition of upstream metabolites of CYP51 (like FPP). However, the phenotype of the mGDH-deficient T. cruzi amastigotes does not support the hypothesis that increased glutamate flux is responsible for sensitization to ketoconazole. Specifically, one would expect that the mutant line should be more resistant to ketoconazole in the presence of glutamine, which is not observed. The authors should reconcile these results with their model. It was noted that T. cruzi may express another active GDH which could compensate for loss of the deleted GDH, or alternatively that the GDH reaction might be by-passed by transaminase reactions. Further experiments may be needed to define the role of these GDH enzymes in ketoconazole sensitivity (including testing the effect of GDH overexpression if effective inhibition of glutamate catabolism is not possible), or these data removed from the manuscript. Along the same lines, further discussion on the phenotype of the cIDH mutant (what was expected and what was observed) is warranted.

Data for the ΔmGDH *T. cruzi* mutants was included in the original manuscript to illustrate the connection between parasite glutamine metabolism (as opposed to host) and the mechanism of amastigote protection from azole-mediated death in the absence of glutamine. However, we agree with the reviewers that the specific nature of this interaction is not yet understood and we have removed this data. We will continue to pursue investigation regarding how the parasite metabolizes glutamine and the implications for protection from azoles.

2) Further explanation of the 13C-glutamine labelling experiments is needed (Figure 3B). The major sterol isotopomers detected in the GC-MS analyses contain +1 and +2 13C-atoms. One would expect +2 or higher isotopomers (i.e. +4, +6 etc) if glutamine is used to synthesize acetyl-CoA via citrate (either by oxidative or reductive pathways). How do the authors reconcile the observed labelling of the sterols and/or have they considered whether fatty acid b-oxidation might also be contributing to elevated acetyl-CoA and sterol synthesis under glucose limiting conditions ?

We agree that if glutamine is incorporated into parasite sterols by similar mechanisms to other systems the +C numbers should be higher than we observe. It is likely that the parasite is using a non-canonical and yet undescribed pathway for this process and the potential role of FA b-oxidation in conditions of nutrient restriction is something we will consider in future. For the 13C analysis the parasites are in glucose, glutamine and FBS (2%) replete conditions. It is possible that in glutamine deplete conditions other metabolic process are altered and indirectly changing the dynamics of sterol synthesis important for the phenotypes we observe. One example of how these possibilities have re-framed the context of these presented data is shown in the sixth paragraph of the Discussion.

3) The evidence for the accumulation of 14-methylated sterol intermediates after ketoconazole treatment is not clear (Figure 3). Specifically, the authors should address the following issues.• Is there any verification to show amastigote purity?

1) We have previously used electron microscopy and western blot for specific host organelles to assess the purity of out isolated amastigote preparations (PMID: 29281741). This is now reflected more clearly in the revised manuscript (subsection “Glutamine-derived carbons are incorporated into amastigote sterols”).

2) Additionally, we analyzed preparation of uninfected cultures for GC-MS to measure purify of our sterol preparations. When uninfected host cells were prepared alongside infected cultures, sterols (e.g. lanosterol, isomers of ergosterol) associated with amastigotes sterol synthesis were not detectable by GC-MS. Trace amounts (<1%) of host derived cholesterol were detectable in preparations from uninfected cultures.

• Figure 3A: the TLC shows a loss of ergosterol/ergosterol isomer after ketoconazole treatment but no accumulation of CYP51 substrates is detected. Is the loading equal?

The loading in Figure 3A is not equal (D2: 4.7e7 and D2+ketoconazole: 2.1e7 amastigotes per lane). Additionally, we cannot control for variable extraction efficiencies prior to analysis by TLC, Figure 3A. We also cannot say that the band running at the same height as the lanosterol standard is composed of a single or multiple species. Therefore, we cannot definitively show accumulation of lanosterol/eburicol with Figure 3A alone. As a result, for clarity we have removed the TLC since the same conclusions can be made, but more precisely with the 13C data alone.

• Figure 3B: it seems that cholesterol is the most abundance sterol species in amastigotes based on natural sterol %, but it is not consistent with Figure 3C.

Cholesterol is the major sterol species in Figure 3A and Figure 3C (Figure 2A, B in updated version). In Figure 3C (now Figure 2A) the major peak (RT 12.98) is the ISTD that is used to normalize between samples. The “natural sterol %” is a metric that reports the percentage within a given species that have 13C incorporation. In this case the % for cholesterol in Figure 3B (revised Figure 2B) reflects that no 13C is detected in cholesterol, not that it is the most predominant species in the sample. It should also be reiterated that the biological (ISTD normalized) variability we see in cholesterol is higher than in amastigote sterols (see modified Figure 2—figure supplement 2) and consequently we cannot confidently quantify cholesterol associated with isolated intracellular amastigotes but we can determine that no 13C is incorporated into the cholesterol that is detectable.

• Resolution for Figure 3B-C is insufficient.

The data in this figure has been enlarged and saved at a higher resolution. In the modified manuscript these data can be found in Figure 2.

• Figure 3C: what is the most dominant peak at 12.8-13.0 in all samples? Is it lanosterol or ergosterol isomer?

In untreated (without ketoconazole) the dominant peak normalized to the ISTD is lanosterol (RT 13.04).

4) Figure 4: while the presented data is compelling, evidence is missing that these isoprenoid precursors are indeed affecting sterol production, rather than acting via a tertiary mechanism. Such evidence should be provided, or conclusions in text tempered.

We agree that this evidence does not directly show that isoprenoid precursors are directly incorporated into sterols. We have added statements to the text (Discussion) that reflect this reality.

We have included new data using BPTES, a small molecule inhibitor of human glutaminase GLS1, that was used to block the conversion of glutamine to glutamate in the host cell. BPTES does not inhibit the ability of isolated *T. cruzi*amastigotes to utilize glutamine for respiration (Figure 3—figure supplement 1A) indicating specificity for host glutaminase. We show that BPTES treatment of infected cells increased the incorporation of 13C-glutamine into amastigote sterols and resulted in an overall increase in amastigote sterol abundance (Figure 3). This result is consistent with a mechanism of increased glutamine availability for the intracellular parasite when glutamine metabolism was blocked in the host cell. Importantly, BPTES treatment also sensitized amastigotes to ketoconazole in the absence of supplemental glutamine, suggesting a correlation between flux into the sterol synthesis pathway and azole sensitivity. However, we cannot rule out additional glutamine fueled pathways in this process as acknowledged in the text (Discussion).

We are also unable to address the influence of host cholesterol scavenging and how this process may influence endogenous sterol synthesis in the parasite (Discussion).

We have also modified the text throughout to temper the conclusions.